



# A Norwegian Approach to Downscaling

Rasmus E. Benestad[1]

[1]The Norwegian Meteorological Institute, Henrik Mohns plass 1, 0313 Oslo, Norway

**Correspondence:** R.E. Benestad (rasmus.benestad@met.no)

**Abstract.** A description of a comprehensive geoscientific downscaling model strategy is presented outlining an approach that has evolved over the last 20 years, together with an explanation for its development, its technical aspects, and evaluation scheme. This effort has resulted in an open-source and free R-based tool, 'esd', for the benefit of sharing and improving the reproducibility of the downscaling results. Furthermore, a set of new metrics was developed as an integral part of the

downscaling approach which assesses model performance with an emphasis on regional information for society (RifS). These metrics involve novel ways of comparing model results with observational data and have been developed for downscaling large multi-model global climate model ensembles. A literature search suggests that this comprehensive downscaling strategy and evaluation scheme are not widely used within the downscaling community. In addition, a new convention for storing large datasets of ensemble results that provides fast access to information and drastically saves data volume is explained.

## 1 Introduction

Global warming and climate change influence both nature and society (IPCC, 2013). The climatic conditions to which human civilisations have been adapted over centuries and millennia are no longer the same, implying new weather-related hazards, risks and opportunities (Field et al., 2012). We must adapt in order to cope with these changes. But climate change adaptation

needs reliable information about the local climate and how it changes, in addition to what weather-related risks are important, and typically how they affect the society. There is also a degree of urgency in our need to adapt to climate change and hence establish a robust system for providing climate related regional information for society. For climate change adaptation it is important to take note that there are different approaches to producing local climate information, known as 'downscaling' (Takayabu et al., 2015). One approach is *dynamical downscaling*, based on regional climate models (RCMs), while *empirical-statistical*

*downscaling* (ESD) uses historical observations and statistical techniques, and *quasi-dynamical downscaling* approach is used to estimate orographic precipitation (Gutman et al., 2012; Barstad and Smith, 2005). In addition, *hybrid downscaling* involves both RCMs and ESD (Erlandsen et al., 2020).





## 2 Regional information for society (RifS)

### 2.1 Main differences to the mainstream

Long experience with both downscaling, and working with impacts and society, has shaped the typical Norwegian strategy for downscaling. The description of this experience serves as a stock-taking on where the science stands on downscaling in Norway, and one objective is to explain *why* this downscaling approach has 'strayed off' the mainstream path. One major difference has been that Norwegian downscaling projects typically have involved a combination of both RCMs and ESD (Hanssen-Bauer et al., 2003; Benestad and Haugen, 2007), a decision that was made already in 1998 in a national research project 'RegClim'[1]

that involved major Norwegian research groups, and that has continued until recent projects funded by the Norwegian Research Council such as 'CixPAG'[2] and 'R-cubed'[3] which also included several Norwegian downscaling communities. The scientific background for the Norwegian climate services furthermore builds on both RCMs and ESD (Hanssen-Bauer et al., 2009). Another major difference involves the way the ESD has evolved and diverged from common approaches within the international community, as illustrated in the schematic presented in Figure 1. The discussion henceforth will mainly elaborate on the

different aspects regarding ESD, as a literary search and common protocols suggest that the merits of the developments within ESD have rarely been utilised outside Norway. Furthermore, the work with RCMs in Norway has followed the main path outlined by CORDEX (Jacob et al., 2020; Vautard et al., 2014), involving RCMs such as HIRHAM (Køltzow, 2007; Køltzow et al., 2008), HCLIM (Belušić et al., 2020), and COSMO-CLM (Dobler et al., 2016). Our work with weather generators (Mezghani et al, in prep.) has also follow similar strategies as the rest of the downscaling community. This paper mainly

discusses our different choices within ESD, but in the end, we synthesise the results from both RCMs and ESD in a combined downscaling approach.

### 2.2 Reasons for why a different approaches

The following discussion will present arguments for *why* different choices were made regarding ESD and *how* they were implemented. The details about exactly *what* was done are in the cited references. Moreover, the objective of this paper is to

explain, in a nutshell, why and how the ESD strategy in Norway diverged from the mainstream. Our strategy has been motivated by the need to deliver information for climate change adaptation, in addition to being based on considerations involving physics, statistical theory and linear algebra. For example, the results from ESD has provided input to the Norwegian climate services and has been used in impact studies that include energy production, ecological response, agriculture, and geohazards.

Here the emphasis on what downscaling means differs to that of CORDEX in a subtle way. According to the CORDEX

website, it is defined as "... *Regional Climate Models (RCM) and Empirical Statistical Downscaling (ESD), applied over a limited area and driven by GCMs can provide information on much smaller scales supporting* ..."[4]. In the present context, the definition of downscaling is *the procedure of adding new relevant and reliable information to global climate model (GCM)*

---

[1]https://projects.met.no/regclim/index_en.html

[2]https://cicero.oslo.no/no/posts/prosjekter/cixpag-interaction-of-climate-extremes-air-pollution-and-agro-ecosystems

[3]https://www.norceresearch.no/prosjekter/relevant-reliable-and-robust-local-scale-climate-projections-for-norway

[4]https://cordex.org/about/what-is-regional-downscaling/





*results, such as how local response depends on the large-scale conditions that GCMs are able to reproduce, or how local geographical conditions play a role.* The subtle difference in emphasis is on the source where this information comes from:
'provide information' as opposed to 'adding new information' with a clear connection to the dependency to large-scale aspects that GCMs can reproduce. The source of additional information in ESD is the historical data and mathematical theory concerning their statistical properties (hence 'empirical-statistical'). The latter take on downscaling also highlights the difference between downscaling and bias-adjustment, which is that the former involves making use of large-scale features that GCMs are able to reproduce. Bias-adjustment doesn't involve the dependency between spatial scales (Gudmundsson et al., 2012), and some will say it gives the right answers for the wrong reasons. It is often required for correcting systematic biases in RCM results, and it is likely that such biases are due to physical inconsistencies. For instance, the RCMs are often physically inconsistent with respect to the driving GCMs, which may involve different outgoing longwave radiation (OLR) aggregated over the same atmospheric volume (Erlandsen et al., 2020), which we also should expect due to different rain patterns (Tselioudis et al., 2012) and clouds that imply differences in the vertical energy flow. There may also be different humidity content in the same air volume embedded by the RCM and the GCM, different parameterisation schemes, and usually different accounts of surface processes and aerosols. One question is whether such mismatches affect the way RCMs simulate climate change, which is one reason for why we need to combine both ESD and RCMs. ESD has other types of shortcomings that are independent to those of RCMs. Estrada et al. (2013) raised a number of concerns about the statistical adequacy in traditional ESD, a condition that justifies their relevance for projecting future scenarios. Their criticism also affected our early downscaling strategy before we changed it.

## 2.3 Different choices

The first step that the Norwegian ESD efforts took in a different direction to the rest of international community happened in 2001. It broke with the common perception that ESD involves the three categories 'Perfect Prognosis' (PP), 'Model Output Statistics' (MOS) and weather generators (WGs) (Maraun et al., 2015) which turned out to be an insufficient classification, as we adopted a hybrid MOS-PP approach involving '*common EOFs*' (Benestad, 2001). A reason for this step was that it adds a layer of quality control, and in retrospect, this technique is both elegant and a superior way of dealing with predictors, compared to traditional PP (Benestad, 2001). Time and time again, it has demonstrated merit (Benestad et al., 2002; Benestad, 2004; Benestad et al., 2005; Isaksen et al., 2007; Etzelmüller et al., 2011; Benestad, 2011; Førland et al., 2011; Hansen et al., 2014; Benestad and Mezghani, 2015; Benestad et al., 2015a; Mtongori et al., 2016; Benestad et al., 2016; Mezghani et al., 2017; Benestad et al., 2018; Mezghani et al., 2019; Parding et al., 2019; Erlandsen et al., 2020). In spite of the success with utilising common EOFs, a Google scholar search on '"common EOFs" downscaling' only had 63 hits (of which about 40 referred to our own work), despite more than 20 years since they first were introduced in ESD and the widespread need for climate change adaptation and downscaled results. This surprising results suggests that they are not appreciated, and this is supported by the fact that common EOFs are not mentioned in text books such as Maraun and Widmann (2018).

One motivation for using common EOFs was to take into account the models' minimum skillful scale (Takayabu et al., 2015; Huth and Kyselý, 2000) and ensure that the same spatio-temporal covariance structure, connecting the predictands





and predictors during calibration, also are utilised for making projections. Often with PP, it's unclear how the results from calibration based on reanalyses are applied to GCMs, and different ways of matching the two can have a profound effect on the results (Benestad, 2001; Estrada et al., 2013). The use of common EOFs requires that both reanalysis and GCM data are

included in the calibration of the downscaling models, hence the reason why they involve a hybrid PP-MOS type approach. The PP aspect involves using reanalysis for calibration, whereas the MOS aspect is using the GCM's covariance structure as part of the calibration. Common EOFs also work best for simple predictors and are well-suited for downscaling large multi-model global climate model ensembles, and common-EOF-based ESD has been applied to the various generations of coupled model inter-comparison project (CMIP) generations (Benestad, 2004; Benestad et al., 2005; Benestad, 2011; Benestad et al., 2016).

Sets of several types of predictors is challenging because the GCMs must reproduce the covariance between them as they appear in the training data. For instance, downscaling of storm track densities suggests that different aspects of in the GCMs may change differently (Parding et al., 2019). Hence, ESD strategies based on several fields need to ensure that the GCMs manage to simulate a consistent change between them so that ESD is not biased by one changing in an inconsistent manner to the others. This type of testing is not commonplace and not among the usual set of quality metrics (Gutiérrez et al., 2018).

## 2.4 "Downscaling climate" approach

If the use of multiple types of predictors is difficult to get right with GCM simulations, then what is the best strategy for downscaling? A simpler and more straight-forward approach is to relate changes in the statistical properties of temperature and precipitation to single-variable predictors. It turns out that statistical properties are surprisingly predictable, are often strongly connected to single predictor variables, and that predicting statistical properties tends to be more robust than predicting

individual outcomes (Benestad and Mezghani, 2015; Benestad et al., 2016). Climate can be defined as weather statistics, which in some circumstances can be quantified in terms of a probability density function (pdf).

The traditional "downscaling weather" (henceforth the "weather approach") aims to estimate the state for each time step of the predictand in order to obtain a time series, and both traditional ESD methods such as the analog model and RCMs belong to this category. By "downscaling climate" (henceforth the "climate approach"), on the other hand, the objective is to estimate

the parameters of pdfs directly (Figure 2). To use the former for estimating a climate change, we would first have to derive a time series, and then use it to estimate a pdf in a less robust way than the climate approach. If we need a time series from the latter, then we need to combine it with conditional WG, however. There are some instances when a time series is needed (e.g. in hydrological impact models), but it is more common that decision-makers want a probability, a threshold value or intensity-duration-frequency (IDF) curves.

Our ESD approach took a second step away from traditional ESD took place in 2007 and involved downscaling probability density functions (pdfs) rather than day-by-day fields (Benestad, 2007; Benestad and Mezghani, 2015; Benestad, 2016), a move that was inspired by Pryor et al. (2005) and dialogues with statisticians. To downscale pdfs, the daily temperature is approximately represented by $T_{2m} \sim N(\mu_T, \sigma_T^2)$, where $\mu_T$ is the seasonal mean temperature and $\sigma_T$ is the seasonal standard deviation. For daily precipitation, we use an expression for the probability of amount exceeding a threshold of $x$ mm: $Pr(X >$

$x) = f_w \exp(-x/\mu)$, where $f_w$ is the wet-day frequency and $\mu$ is the wet-day mean precipitation (this expression is strictly





speaking not a pdf). We use seasonal aggregation to get sufficient samples (90 days) in order to obtain good estimates of the statistics. Since we downscale the parameters of the pdfs and the probability expressions, i.e. $[\mu_t, \sigma_T, f_w, \mu]$, we tend to use multiple regression because parameters aggregated over seasonal scales tend to approximately follow the normal distribution according to the central limit theorem. Our philosophy has been to keep the analysis simple and elegant (mathematical). This
strategy has been successful, as we will see in the example below.

### 2.5    Common protocols concerning downscaling

Since mainstream ESD approach generally adopted by the international community (CORDEX-ESD[5] and COST-Value[6]) tends to follow traditional classifications and belong to the weather approach, the adopted protocols do not accommodate for neither the hybrid PP-MOS nor the climate downscaling approach. For instance, the proposed protocol is based on comparing how well
different statistical methods perform for the weather approach, given ERAINT (Dee et al., 2011) as predictor for the period 1979–2013 and involving cross-validation. But, they are not suitable for testing the models in a real setting when applied to GCMs, such as in the context of the common EOF-based PP-MOS hybrid downscaling. A relevant way of evaluating the downscaling, on the other hand, may have a stronger emphasis of how reliable the downscaled results from the GCMs are (Benestad et al., 2016), and to lesser extend for one given method in ideal conditions and a controlled environment. Gutiérrez
et al. (2018) presented a comparison for a similar protocol used in the project COST-Value, which also involved validation aspects such as marginal, temporal, extremes, spatial, process-based aspects (Maraun et al., 2019, 2015). While cross-validation always is important, there are several metrics proposed for validation by COST-Value that are not relevant for the Norwegian approach to ESD, as they don't address the aspects such as changes in pdf parameters. Another example is the spatial metric proposed through the COST-Value project, which implies that the use of principal component analysis (PCA) as predictands
in ESD is not wide-spread. A third step away from the traditional mainstream ESD too place in 2015 and involved using PCA to represent the predictands consisting of estimates of pdf parameters aggregated from daily data (Benestad et al., 2015a). We use PCA to represent the predictands because they inherently ensure the same spatial covariance as seen in the observations, in addition to reorganising the data to emphasise the large-scale variability (Benestad et al., 2015a).

### 2.6    ESD is suitable for downscaling extremes

Contrary to common perceptions (Solomon et al., 2007, 11.10.1.3), ESD based on the climate approach is suitable for making inferences about extremes, and probably more so than RCMs. In addition to downscaling the probability of heavy rainfall, as presented above, we have proposed ways of downscaling pdfs to study how the duration and frequency of heatwaves is affected by climate change (Benestad et al., 2018). This strategy is inspired by dialogues with statisticians, and is based on

---

[5]https://cordex.org/domains/cordex-esd/

[6]http://www.value-cost.eu/





the geometric distribution which describes the probability distribution of the number $X$ of Bernoulli trials needed to get one
success (Wilks, 1995). The probability $Pr(X)$ can be defined according to

$$Pr(X = k) = (1 - p)^{(k-1)}p \ \forall \ k = \{1, 2, 3, ...\}, \tag{1}$$

where $k$ is the number of consecutive days with heat and $p = 1/\overline{d}$ is the probability of no heat on any given day ("success
probability") and is a function of the mean spell duration $\overline{d}$. This approach involves downscaling of parameters for the duration
pdf, and is another example where our approach is incompatible with the standard set of metrics suggested by Maraun et al.
(2015) for evaluating temporal properties of downscaled climate, which consisted of the *median* of spell length and the 90th
percentile of spell length. The median can only be an integer number (or halfway between two integer numbers), and is not
a pure rational number such as the *mean* spell length that is needed to represent the parameters of a pdf. This is not the only
example, as the statistics for consecutive wet and dry days (CWD and CDD) used in CLIMDEX[7] and Donat et al. (2016) is
based on the *maximum number of consecutive days* with 24-hr rainfall above or below a given threshold (1 mm/day). This
use of spell duration statistics has also been used in the IPCC SREX, presenting the change in annual maximum number of
consecutive dry days (Field et al., 2012, Figure SPM.5). None of these metrics account for how the spell statistics can be
connected to a changing pdf. The statistics of maxima is often cluttered and involves a high degree of uncertainty, is not a
rational number, and we believe we can get more robust results by fitting the parameters of a pdf of a probability formula
(Benestad et al., 2019b, 2020). Hence, it's better to use the mean number of consecutive days $\overline{d}$ since it fits better within the
framework of the geometric distribution. To estimate the probability of long-lasting events, the geometric distribution needs
to be combined with the probability of the events occurring (e.g. Bayesian approach where the probability of the event can be
modelled as a Poisson process). Moreover, the mean number of consecutive days both enables the estimation of the probability
of events lasting longer than a given length and is less affected by random sampling fluctuations than the maximum from a
given period.

Our recent efforts involve new ways of downscaling extremes trough ESD, and we explore the potential to use an approx-
imate parametric formula for IDF curves (Benestad et al., 2020) and the shape of the mathematical curves representing IDF
statistics (Parding et al., in prep.). The fact that these curves have a simple shape that can be described by a small set of pa-
rameters that respond to climate change is encouraging. We have also concluded that it is not always necessary to involve a
covariance structure, as ESD also can take aggregated statistics and indices as predictor. A couple of examples are the number
of tropical storms based on warm ocean surface area (Benestad, 2009) and the global area of 24-hr rainfall based on the global
mean temperature (Benestad, 2018). The use of global mean temperature as predictor was inspired by Oldenborg and Ulden
(2003).

---

[7]https://www.climdex.org





## 2.7 Levels of evaluation for ensuring reliability

When it comes to the evaluation of our downscaling efforts, we have concluded that nine levels are needed to address concerns related to both the performance of the individual method and the quality of regional information provided to the society:

1. Cross-validation (Figure 3)

2. Detrending test for stationarity over the calibration period (Figure 3)

3. Common EOFs ensure representative spatio-temporal covariance structure in the GCMs to represent the large scale predictors (Benestad et al., 2016).

4. Test of whether ensemble of downscaled results have trends over the historical period that match the observed one (Figure 4).

5. Test of downscaled ensemble 90% confidence interval spanning the observed interannual variations (Figure 4).

6. Compare RCM results and ESD for corresponding GCM run (Mezghani et al., 2019).

7. Using "pseudo-reality" for method testing (Erlandsen et al., 2020).

8. Diagnostic check for whether the predictor pattern reflects physically plausible link (Figure 3).

9. Check if statistical distribution of parameters is normal and daily values are according to the assumed pdf (Figure 5).

Figure 3 shows an example where we test our ESD method in a PP-mode applied to Nordic winter temperatures represented through a PCA-based framework. In this case the predictor is the December-February mean temperature from the ERA5 reanalysis (Hersbach and Dee, 2016) over the period 1950-2019 (70 years) aggregated to a coarser grid to better match GCMs. The predictand consists of the three leading PCAs of Nordic $\mu_T$ from ECA&D averaged over the same season (Klein Tank et al., 2002)[8]. The figure presents both the traditional cross-validation results (level 1) for the leading mode and the method's ability to capture the long-term trend, hence a test of non-stationarity (level 2) (Huth, 2004). Both these tests are also used when common EOFs are used as predictors.

Figure 4 presents the output of our ESD approach applied to a multi-model CMIP6 GCM ensemble following the SSP370 emission scenarios (Eyring et al., 2016). In this example, we used common EOFs, and hence the hybrid PP-MOS approach. The diagnostics provided include an evaluation of the GCMs' ability to reproduce the spatio-temporal covariance (level 3) to check whether the large-scale predictors simulated by the GCMs are realistic (Hellström and Chen, 2003), a test whether the downscaled historical part of the simulations are consistent with the observed trend (level 4) and whether the downscaled ensemble gives a range of values similar to the observed interannual variability (level 5) (Vrac et al., 2007). We should keep in mind that we shouldn't expect the three leading PCAs to capture all variability at a single stations (in this case, they accounted

---

[8]https://www.ecad.eu/documents/atbd.pdf





for 99.7% of the variance) and the dependency to the large-scales is not expected to account for 100% of the variability (level 5).

We do not present results from level 6 here, but such evaluation in the past has given consistent results between both ESD and RCMs (Mezghani et al., 2019, 2017). This is also the type of information provided in the Norwegian climate services

(Hanssen-Bauer et al., 2009). Evaluation level 7 involves a more general test to explore the merit of the climate approach. We see that the downscaling of wet-day mean precipitation $\mu$ is difficult (still being explored), and that the length of the calibration interval and choice of predictor matter. For temperature with large-scale temperature as predictor, we need at least 70 years for obtaining robust trend reproduction. This means that ERA5 1950–2019 is a good choice.

Our evaluation also involves comparing different emission scenarios to explore the sensitivities to different future emission

scenarios. In addition, we analyse the observations to learn what aspects and variables are sensitive and what are insensitive to different physical conditions. We carry out sensitivity analyses to explore the general impact of changing physical conditions on the predictand in terms of the mean seasonal cycle, past trends and the sensitivity to geographical location (Benestad et al., 2016; Thomas et al., 2007; New et al., 2006). In addition, we make a manual visual inspection of the predictor pattern when we test out different predictors (level 8), with an expectation that the patterns should make sense physically (Figure 3). We also

need to test whether the presumed probabilistic model is representative for the observations with standard statistical tests, e.g. the that normal distribution is representative of daily temperature ('qqnorm'), and whether the parameters for seasonal statistics are approximately normally distributed, as we would expect from the central limit theorem (level 9; Figure 5).

The climate approach is in a better position than the mainstream weather approach to accommodate for concerns with probabilistic assumptions (Estrada et al., 2013). For instance, the requirement of independence is more readily fulfilled because there

is little predictability from a season one year to the same season the next year. Also using detrended data for calibration alleviates some of concerns about the data not being independence and identically distributed (iid), and downscaling the parameters rather than the time series is also designed to account for the fact that the statistics may change over time. Finally, we apply gridding to the ESD results thought kriging with elevation as a covariate (Benestad et al., 2016), which also adds information and can potentially level out single sites with spurious results (Figure 5). We have shared our methods as an R-package 'esd'

that is freely available from https://github.com/metno/esd and comes with examples, explanations (e.g. on its wiki-page) and documentation (Benestad et al., 2015b). Several of our papers have included R markdown scripts with the recipe of the specific analyses as supporting material to improve the reproducibility of our results.

## 2.8 Strategy for storing large volumes of multi-model ensemble ESD output

Finally, our strategy also has enabled us to produce the large downscaled multi-model ensembles, based on the entire CMIP3/5/6

multi-model ensembles (Benestad, 2002, 2004, 2011; Benestad et al., 2016). A recent publication on ESD included 254 CMIP5 runs (RCPs 2.6, 4.5, 8.5) (Benestad et al., 2016), and for the forthcoming CMIP6 runs, we plan to do more than 1000. The test setup presented in Figures 4 and 6 represents 151 downscaled SSP370 runs for the one combination of scenario, season and parameter for all of the Nordic countries shown. This test was completed in 48 minutes on Centos 7 running on an Intel(R) Xeon(R) CPU E5-2660 v3 at 2.60GHz, highlighting the extreme computational efficiency of the Norwegian approach





to ESD, and it can be carried out in parallel for all combinations of scenario-season-parameter and be completed within days for the entire set of CMIP6 multi-model GCM ensembles for both temperature and precipitation statistics. It is crucial to use large ensembles for providing regional information for society in order to avoid misrepresentation due to the *"Law of small numbers"* (Kahneman, 2012) and the strong presence of stochastic regional variability on decadal scales (Deser et al., 2012). Mezghani et al. (2019) demonstrated that the Euro-CORDEX RCM ensemble gives a biased sample of the CMIP5 simulations

over Poland and underestimates the future warming. However, it is difficult to handle the shear data volume of large ensembles of downscaled results. We deal with vast volumes of data containing high-resolution maps from extensive multi-model ensembles by proposing a new and more suitable format for storing large datasets, which differs radically from the traditional data archives with netCDF files following the 'CF' convention (Benestad et al., 2017a). Our method is far more efficient than individual netCDF files for each ensemble member, which is demonstrated in https://esdlab.met.no/BarentsAtlas/. Furthermore, it

facilitates the extraction of information in a faster and more computationally efficient way than traditional archives by far, such as ensemble mean, ensemble spread, individual models, percentiles, time series, single years or trend analysis. Figure 6 shows an example of the 95-percentile for 151 simulations from CMIP6 SSP370 for 2050 December–January $\mu_T$ that was extracted in seconds from the large volume of data. In our case, it accommodates for both field objects as well as station-based time series and is specially well-suited for seasonally aggregated parameters for pdfs of daily temperature or precipitation since almost all

of their variance can be accounted for by a small set of leading PCAs/EOFs.

### 2.9 Robust climate change adaptation from ESD

It is important to ask exactly what information is used and exactly how is it used, We may take it for granted, but it is crucial use the right information in correct way. For instance, relying on results from a single dynamical downscaling exercise with one simulation by an RCM and a GCM is unwise because if we chose another GCM simulation to downscale, we

would get a different answer. In fact, basing information for society on a small set of driving GCMs ($n < 30$) is likely to give misleading results subject to stochastic fluctuations (Mezghani et al., 2019). Even the traditional Euro-CORDEX RCM ensemble is a case of not using the right information in correct way, as all RCMs in the ensemble may have systematic biases with the same sign because of common physical inconsistencies in terms of OLR and common shortcomings in coupling with surface/ocean/lakes or treatment of aerosols. Furthermore, average rainfall over a grid area of $10 \times 10 km^2$ is expected

to have different statistical properties to rain gauges with cross-sections of the order of centimetres. Hence, their results do strictly not represent the same aspects as those observed. RCMs nevertheless have great value in the context of experiments and studies of how different phenomena respond to different boundary conditions, such as convection or how heatwaves are exacerbated by low soil moisture. In this case, they can add value when used to address specific research questions or test hypotheses concerning regional climatic aspects. Techniques from ESD can be used to analyse the output of RCMs in terms

of "pseudo-reality" and contribute to enhanced understanding of what the models do. Nevertheless, both RCMs and ESD are needed for climate change adaptation, and combination of results from both RCMs and ESD means adding more information to the equation and hence provide an enhanced understanding of how robust the results are and what measure of uncertainties is present.





So how is information from ESD presented here used by society? The example shown in Figure 5 can involve using
$N(\mu_T, \sigma_T)$ to estimate approximately how many days with freezing temperatures we can expect in a warm winter at around
2050 as long as the threshold is not too far out in the tails of the distributions. There is, however, a valid concern regarding
whether a normal distribution is sufficient for the representation of daily temperature statistics (Figure 5). For more extreme
cases, it may nevertheless be better to downscale the number of events more directly by assuming a Poisson-type family and
generalised linear models (GLMs). Furthermore, similar maps of the seasonal wet-day frequency and wet-day mean precipi-
tation can be used to estimate the probability for heavy precipitation, e.g. above 50 mm/day (Benestad et al., 2019b), and the
same parameters may furthermore be used to provide approximate estimates for IDF curves (Benestad et al., 2020). These
results are robust because they are derived from large multi-model ensembles of GCM simulations with added information
about how the local climatic response depends on their simulated large-scale features and the effect of systematic geographic
influence. Such results are not sensitive to the regional stochastic decadal variability nor the replacement of a few GCM sim-
ulations. Furthermore, the results have been subject of the said nine levels of evaluation. A lingering question, however, is
whether a multi-model ensemble provides an unbiased representation of probabilities for the outcome (Benestad et al., 2017b).
The evaluation of the historic trends and interannual variability will provide some indication about how well the downscaled
ensemble represents the observations.

ESD can provide more information than the said example and we have only started to explore the potential for downscaling
the mean duration of dry spells (droughts), heat waves and cold outbreaks. Other aspects that may be relevant for society
is the area of the events, which can be derived from gridded data (Lussana et al., 2019), and perhaps radar or satellites, for
instance through indicators capturing a combination of area, duration and intensity. Also extremes can be both downscaled
as a number of phenomena or derived indirectly through the parameters of an appropriate pdf. The least suitable method is
to derive their statistics from time series, as in traditional ESD (Snäll et al., 2009) or bias-corrected dynamical downscaling.
In Snäll et al. (2009) we used ESD to downscale daily maximum temperature, but it required an adjustment due to reduced
variance and it followed the traditional approach subject to some of the criticism presented by Estrada et al. (2013). A better
approach would be to downscale the number of hot days or the probability of heatwaves (Benestad et al., 2018). One example
of downscaling the phenomena is the storm density (Parding et al., 2019) and typical number of storms within a region per
season. Proper use of ESD, however, requires a deep understanding of what type of answers that are sought and a combination
of geography, physics and statistics to find a tailored method. The question how to extract the right type of information for the
right use merits more wide-spread discussions involving the whole geoscientific community on downscaling. Another point is
that regional information for society and climate change adaptation should not only involve the linear downscaling approach,
but also "bottom-up" and sensitivity analysis (Mtongori et al., 2015) and stress testing (Benestad et al., 2019a). The latter may
not be scientific, but may still provide useful input to adaptation strategies. The "bottom-up" approach has been championed
by Pielke Sr. and Wilby (2012).





# 3 Conclusions

In summary, we propose nine levels for validation when we provide regional information for society, which to some extent takes into account skepticism expressed against downscaling (Pielke Sr. and Wilby, 2012) and an observation that researchers "row over whether regional projections are based on sound science" (Rosen, 2010). Different strengths and shortcomings associated
with RCMs and ESD is the reason why it's important to bring them together (Hellström et al., 2001). For instance, both ESD and RCM assume stationarity, the former though downscaling dependencies and the latter though upscaling of unresolved processes. Moreover, they add information on regional scales based on different sources, which also are independent on each other. Furthermore, they both complement and support each other. RCMs offer descriptions that are not available from ESD, such as fluxes and a complete coverage.

The reason why downscaling in Norway typically diverged from the main path was to ensure robust and reliable results for society, and how we did it was, in short, (1) to introduce a hybrid PP-MOS scheme, (2) use PCA to represent our predictors, (3) adopt the climate approach, (4) adopt nine levels of evaluation, (5) make use of really large multi-model ensembles, (6) use pseudo-reality, and (7) bring together ESD and RCM results. We mainly downscaled temperature and precipitation statistics, but have also downscaled storm track density (Parding et al., 2019). The Norwegian approach has been tailored for regional
information for society and climate change adaptation, something that is reflected in our nine-point evaluation scheme and in the seven said points.

The divide between the different approaches highlights a need for more extensive and deeper debates to ensure best use of downscaling for climate change adaptation and tackle the "practition-er's dilemma" (Barsugli et al., 2013). Important questions include *why* and *why not* we should follow different practices ('*hows*'). The protocols for evaluation may also benefit from
further revisions, reflecting stronger commitment to bring together ESD and RCM and avoid the "silo thinking" dominating the present. There have been some attempts to bring together RCMs and ESD (Jacob et al., 2020), but the development towards a more unified approach is too slow, considering the speed of global warming.

*Code and data availability.* A set of demonstration scripts and data presented herein are available from https://doi.org/10.6084/m9.figshare.14922837.v1.

*Acknowledgements.* This paper wasn't funded by any external projects, but represents a contribution to general scientific discourse and progress.



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

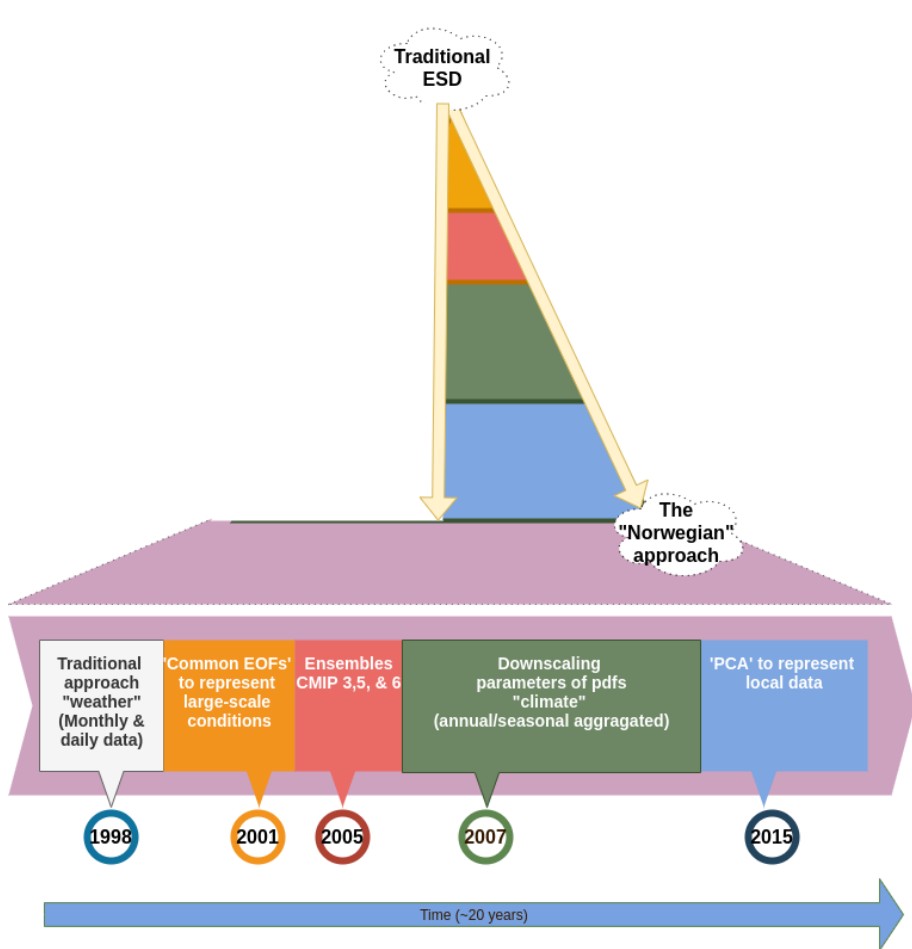

**Figure 1.** Illustration of the evolution of the ESD downscaling strategy, where the upper part illustrates how the downscaling strategy diverged from a more traditional approach adopted in 1998, and the lower part shows a crude timeline.



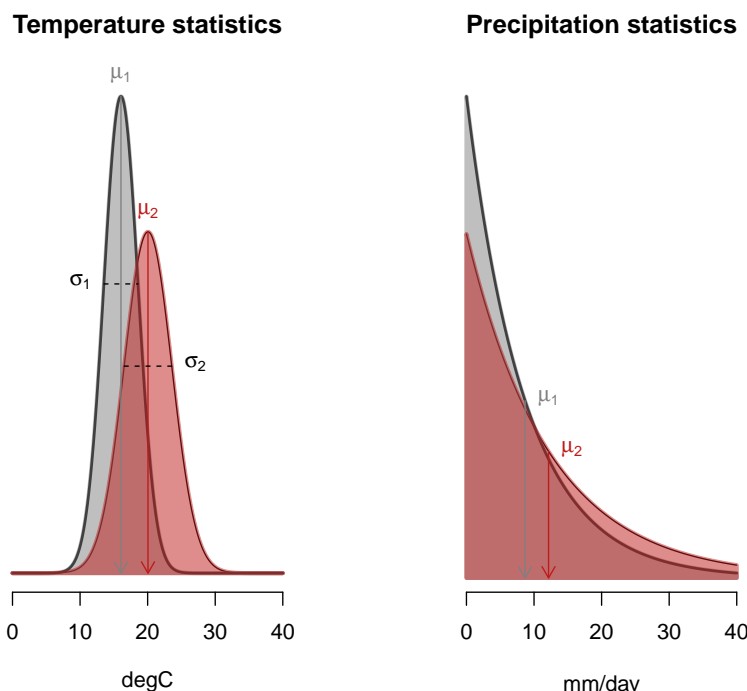

**Figure 2.** Illustration of pdfs and an ideal climate change for daily temperature and wet-day precipitation amounts, where the grey curve represents an original climate and the red curve a new climate. The change in these pdfs are specified in terms of the parameters $\mu$ and $\sigma$.



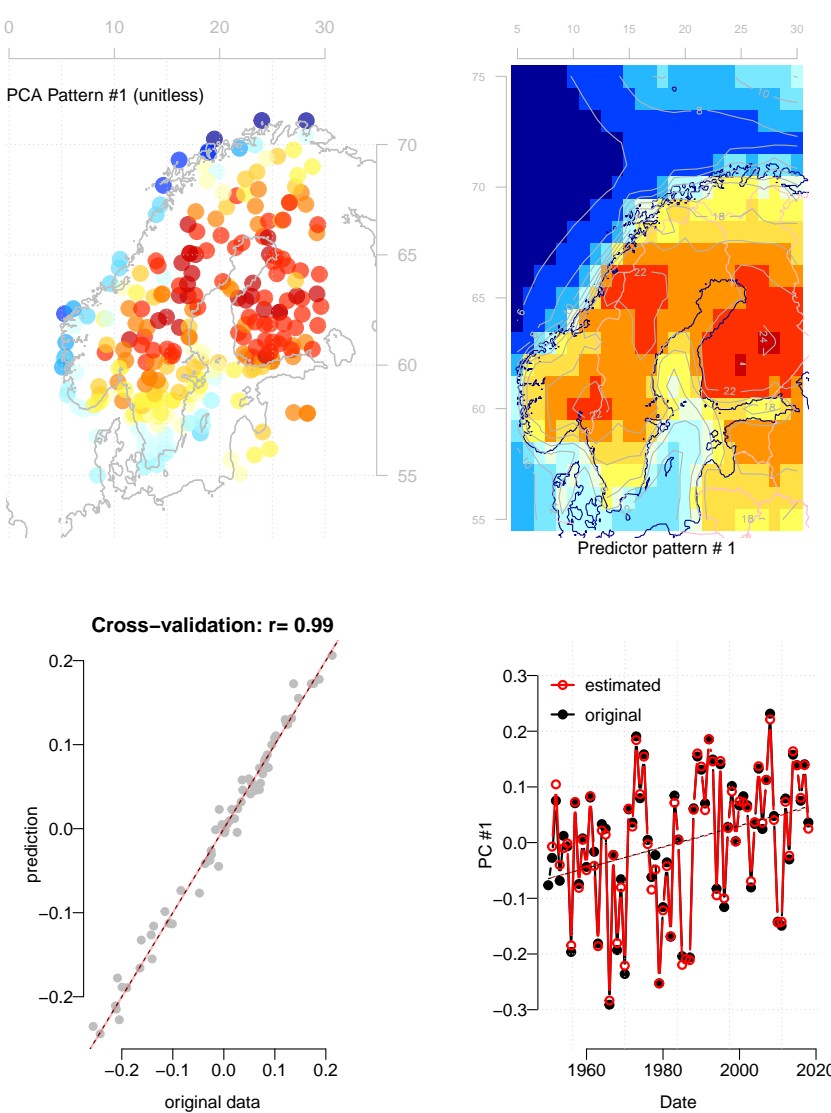

**Figure 3.** Example of of evaluation levels 1–2 for a PCA-based ESD test for December-February $\mu_T$ for 310 locations in the Nordic countries, with ERA5 1950-2020 as predictor for calibration. The upper left panel shows the weights of the leading PCA, upper right panel shows the predictor pattern, the EOFs of the ERA5 temperature weighted by the coefficients from the multiple regression (after aggregation to coarser grid to better match the GCMs). The lower left shows the results of a five-fold cross-validation, and the lower right presents the observed and predicted trends. The prediction of trend involved detrended data for calibration and original data as input to give the output.





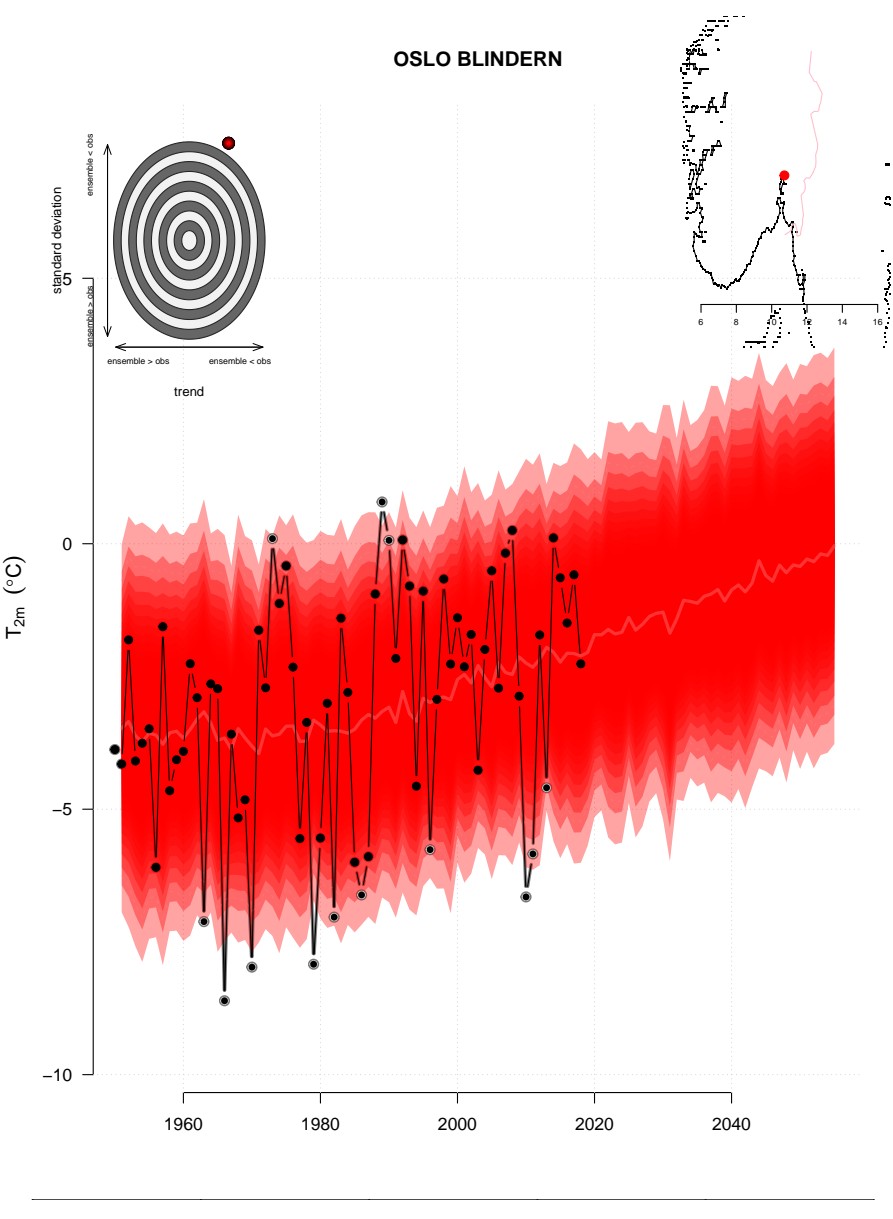

**Figure 4.** Example of evaluation levels 3–5 for downscaling results for December-February $\mu_T$ for Oslo, based on calibration with ERA5 reanalysis, common EOFs, and 3 leading PCA applied to 310 ECA&D time series of daily temperature from the Nordic countries (accounting for 99.7%). Here the downscaled results are based on 151 CMIP6 SSP370 runs with coverage 1951-2050. The trend statistics from downscaled ensemble is consistent with observed trends, but the downscaled interannual variability is somewhat reduced. The cross-validation correlation coefficients for the three PCs and the ensemble of common EOFs associated with each CMIP run were within the ranges 0.98–0.99, 0.73–0.91, and 0.49–0.77 for each respective modes.

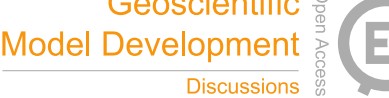

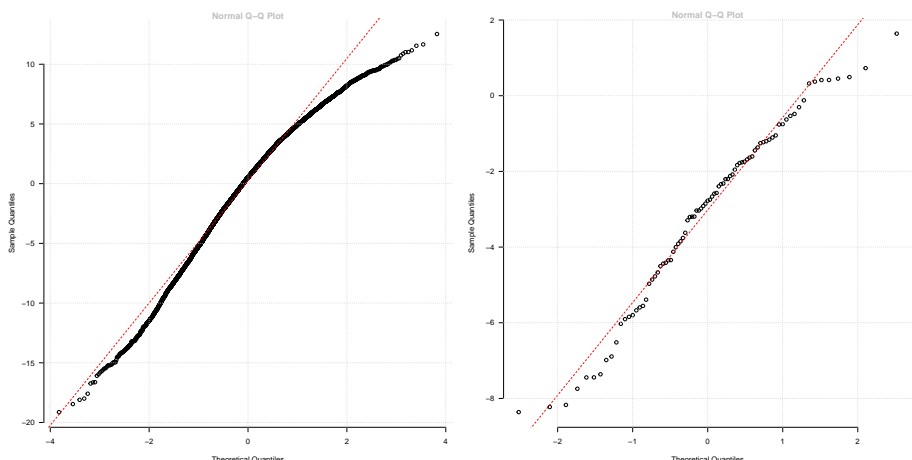

**Figure 5.** Example of evaluation levels 9 for downscaling results for December-February $\mu_T$ for Oslo. The left panel shows how well a normal distribution fits the daily winter temperature whereas the right panel shows a similar test for the parameter $\mu_t$ used for calibration. The fit to normal distribution is not perfect, but nevertheless a useful approximation. The largest divergence is founds near the tails, which is not surprising.



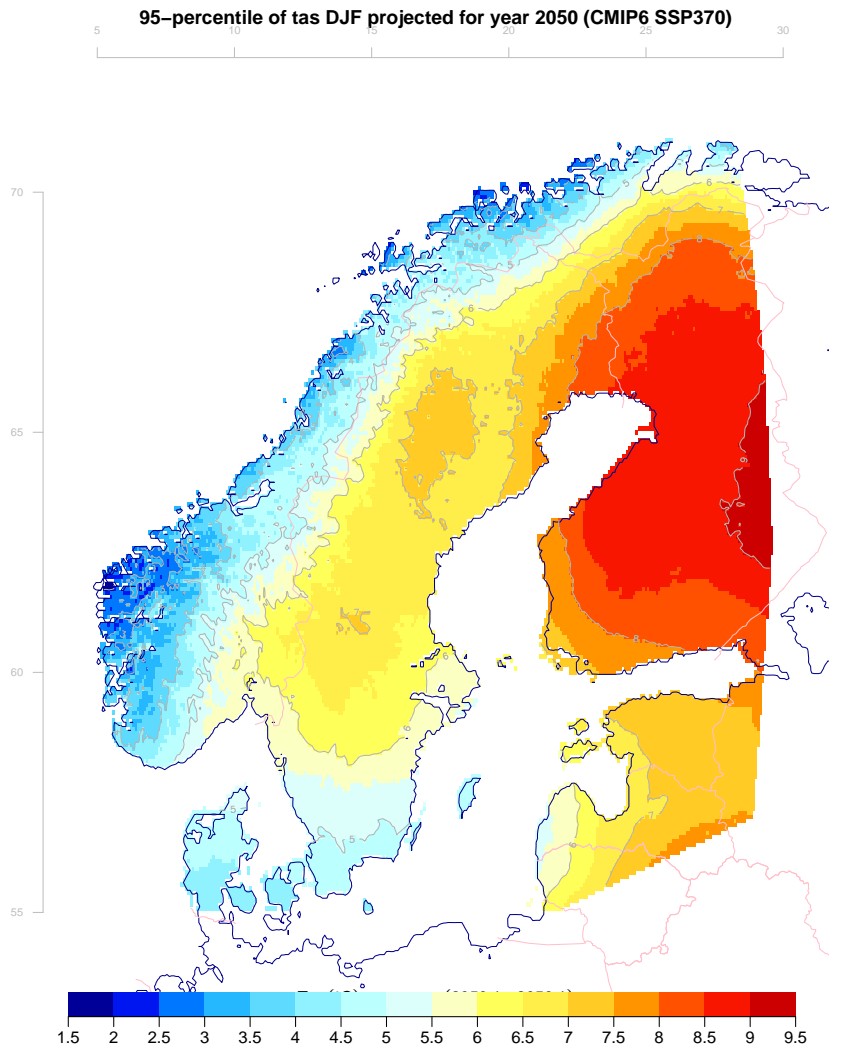

**Figure 6.** Example of ensemble 95-percentile extracted from 151 CMIP6 SSP370 simulations for 2050 December-January $\mu_T$. These results are the same as those presented in Figures 2-6, and all available CMIP runs for the period 1950-2050 were extracted and a kriging with elevation as a covariable was applied to the the PCAs representing the predictand to provide gridded results on an $8 \times 8$ km resolution.