# Peer review of "A Norwegian Approach to Downscaling"

_Geoscientific Model Development, 2021_

## Author Comment (AC1)

**Response_to_gmd-2021-176-RC1**

REB

10/19/2021

**Response**

I want to thank anonymous reviewer #1 (DOI:10.5194/gmd-2021-176-RC1) for a frank and critical take on Benestad (2021). It is extremely important with discussions such as this one, even if we don't agree on the choices or methods. Because the merit of discussions potentially may be degraded by misunderstandings, a risk which can be reduced by including replicable and transparent demonstrations, I will use an R-markdown script here and its output in the PDF format. It will provide a demonstration based on the open-source `esd-package` that is freely available from https://github.com/metno/esd (documentation, help pages and demos are available on its wiki-page).

**'Review' and lack of originality?**

Reviewer #1 reads the paper as "a review of the statistical downscaling strategies followed by the author over the past decades." No. It provides an overview of our strategy, and refers to past work as examples supporting this strategy. I realise that our work to large extent has been ignored by the international downscaling community. There are indeed few other groups that have carried out similar work. This is the reason why I had to cite much of our own research to demonstrate the merit of our strategy. It would be nice to be able to refer to other examples underpinning the discussion of the overall downscaling strategy, but Reviewer #1 did not suggest any papers discussing similar strategies as Benestad (2021). A lack of suitable references can be discerned from the exclusion of other similar work in the chapter on regional modelling (Chapter 10) in the latest IPCC AR6 WG1 report from 2021. For instance, the hybrid PP-MOS type of downscaling models, that provides much of the basis for this downscaling strategy, is not included in Table 10.1 in Doblas-Reyes et al. (2021). Common EOFs are only mentioned briefly once, with a brief reference to Benestad (2011) and then mislabeled as 'PP.' It would be relevant to explain their purpose, but Doblas-Reyes et al. (2021) doesn't. Nor does the report mention them in connection with the discussion on the ability of global models to simulate large-scale indicators of climate change. None of the evaluation strategies for downscaled ensembles mentioned are discussed (they are explained in both Benestad (2021) and Benestad et al. (2016)), despite their natural place in the discussion of the overall performance in a climate change application and performance of the driving climate model ("model fitness for projections"). There is no mention in the IPCC report of the use of PCA to represent predictand and its benefit in terms of preserving the spatial consistency between sites, which naturally would fit in the chapter. These omissions can be interpreted as being exclusive since some of the authors on this chapter should have known about these topics. They also explain the difficulty to find other relevant references that Benestad (2021) could cite in order to support the choices made in the downscaling strategy.

**Dubious aspects?**

Reviewer #1 claims that "reasoning concerning a few technical aspects is dubious" and that "it does not contain original model or methodological descriptions." Both are wrong and are not supported by any demonstration nor accompanied by convincing details. The good news is that most of the methods in Benestad (2021) have been demonstrated in the cited papers and a further demonstration is presented below.

When we work with the data, we can also test the assumption which the example provided shows. It is hopefully possible to resolve many different interpretations by testing claims on real data.

This is the first time we have written up a comprehensive description of our overall downscaling strategy, and I don't think there are many other similar descriptions. The fact that hardly any of this work was cited (and certainly not discussed) in Doblas-Reyes et al. (2021) shows that the work still is original from an international point of view. I will challenge Reviewer #1 to point to any published papers discussing the nine evaluation criteria that were introduced in Benestad (2021). Also, it's difficult to find other examples of downscaling that include the validation of the driving GCMs. A search within Doblas-Reyes et al. (2021) for such examples was unsuccessful. Again, if Reviewer #1 knows of any, I hope (s)he will share this information with us.

**Overview versus a forest of details**

One critical point that Reviewer #1 has on Benestad (2021) is that it "refer[s] to a list of previous papers without offering deeper insight." Again, the manuscript provides an overview of the comprehensive downscaling strategy and provides examples of demonstration to back up the choices taken. Here, the level of details was kept to a minimum to avoid getting lost in the forest of details. The cited papers should be open-access, so readers who want to delve into the details can do so by reading the cited papers. It would be no point of repeating them here, as it would introduce material that is not original - by adding such, the paper would become less original and that would give the criticism of lacking originality a little substance. Again, the purpose of this paper is to provide an overarching view and understanding of the approach to downscaling and why it differs from other similar efforts.

**'Project report?'**

The interpretation that Benestad (2021) 'the manuscript to [Reviewer #1] rather reads like a project report or a report of a laboratory for an external evaluation' is both strange and subjective. It shows that there are very different ways of looking at downscaling and scientific literature. I have used an R-markdown script here to demonstrate what a 'project report or a report of a laboratory' looks more like. It may even seem that Reviewer #1 is trying to exclude scientific contributions (s)he doesn't like.

**The title**

The original title of Benestad (2021) was 'The Norwegian downscaling Strategy,' but after I received some comments from Norwegian colleagues, it was changed to 'A Norwegian downscaling Strategy.' It does of course not reflect all downscaling in Norway, but nevertheless touches a large portion of it over the 23 years since the national RegClim project. Also, this approach has all the time combined both RCMs and ESD, as explained in Benestad (2021), which I have realised is not so common in other countries. And of course, our approach to ESD is very different as explained in the paper.

**The abstract**

The comment from Revewer #1 ("what is the downscaling method based? is it statistical or dynamical ?") reveals our different points of perspectives and different sets of expectations. This paper provides an overview of the comprehensive downscaling strategy that is a level above the choices of a particular downscaling method or whether it is dynamical or statistical. That would be a traditional way of writing a paper, but here we take an innovative and original look on downscaling on a higher level. The paper does present what is stated in the abstract: "...description of a comprehensive geoscientific downscaling model strategy is presented outlining an approach that has evolved over the last 20 years...."

**Long experience**

The work goes back to the end of the 1990s, and there were few downscaling groups then. There were some, but many of the scholars in those have since retired. Also, it predates CORDEX. The point is that the work on downscaling over many years has produced good progress that has not been recognized, as shown above where Doblas-Reyes et al. (2021) is a case in point.

**Stock-taking**

I'm certainly not representative for whole Norway, but have collaborated with largest research organisations involved in downscaling in Norway: CICERO, University of Oslo, NORCE, Norwegian Computing, and NVE. Most of these use RCMs and are well attuned to Euro-CORDEX. The paper focuses on ESD (for which I know of few other activities in Norway) and the combination of ESD and RCM.

**Bias correction**

There may be different views on bias correction, but here I want to emphasise that the merit of downscaling is to use the large-scale aspects that the GCMs are able to reproduce in a skillful way, and information about how local conditions depend on such large-scale conditions, in addition to geography, and use these additional sources of information to get a refined picture. Bias correction involves bias correction. It's legitimate to state such a point of view in a scientific paper, I think, and especially if it's controversial.

**Common EOFs**

The paper does not assume that common EOFs are superior, but cites demonstrations that show that they in fact are superior. The strange thing is that they are not recognized in Doblas-Reyes et al. (2021), but only briefly mentioned once in the passing. This is also part of the motivation behind describing the downscaling strategy that has been adopted in a Norwegian research group with long experience on downscaling. We have understood that the concept of common EOFs perhaps is a bit difficult to understand.

**Outcome**

Thanks for this question - this is explained more carefully in the revised version of the paper: "In this context, individual outcomes can be the temperature or rainfall for a random day or a particular state for a random time step when we deal with a time series."

It is true that climate downscaling is never about the prediction of individual events, and the difference between the two concepts here is in terms of how the models are calibrated: either on a time-step-by-time-step (day-by-day) basis or by estimating the dependency of the pdf parameters on large-scale conditions. This is not more carefully explained in the revised paper.

**Normal distribution according to the central limit theorem**

The reviewer points out that the notion of a normal distribution for aggregated statistics is not generally true, and in a bayesian setting the prior for variance are other types of non-normal distributions. In this case, we discuss only four different types of aggregated statistics: the mean $\overline{T}$, the standard deviation $\sigma$ for temperature, and wet-day mean precipitation $\mu$ and wet-day frequency $f_w$ for 24-hr precipitation. Fortunately, we don't need to downscale the variance. We can also carry out actual tests to see if the distribution of these variables do follow a normal distribution. Below is an example for the standard deviation $\sigma$ that has the closest connection to the variance $\sigma^2$ cited in the criticism:

```
**Test the distribution of variance in temperature**
library(esd)  ## https://github.com/metno/esd
```

```
**Loading required package: ncdf4**
```

```
**Loading required package: zoo**
```

```
##
**Attaching package: 'zoo'**
```

```
**The following objects are masked from 'package:base':**
##
**as.Date, as.Date.numeric**
```

```
**Registered S3 methods overwritten by 'esd':**
**method          from**
**subset.default  base**
**subset.matrix   base**
**subset.zoo      zoo**
```

```
data(ferder)  ## This line fetches 24-hr mean temperature for Færder lighthouse south of Oslo
sigma <- as.4seasons(ferder,FUN='sd')
djf <- subset(sigma,it='djf')
mam <- subset(sigma,it='mam')
jja <- subset(sigma,it='jja')
son <- subset(sigma,it='son')

par(mfcol=c(2,2))
qqnorm(djf,main=expression(paste('DJF qqnorm: ',sigma)))
qqline(djf,lty=2,col='red')
grid()

qqnorm(mam,main=expression(paste('MAM qqnorm: ',sigma)))
qqline(mam,lty=2,col='red')
grid()

qqnorm(jja,main=expression(paste('JJA qqnorm: ',sigma)))
qqline(jja,lty=2,col='red')
grid()

qqnorm(son,main=expression(paste('SON qqnorm: ',sigma)))
qqline(son,lty=2,col='red')
grid()
```

[Figure]

The example here shows that $\sigma$ doesn't follow a normal distribution perfectly, but the normal distribution is nevertheless a reasonable assumption.

It is also explained in Benestad (2021) that it's extremely important to evaluate all assumptions and all results, and also to test the downscaled results for $\sigma$ against observations to evaluate the skill of the ordinary linear model (OLM). Trend analysis over historical times and downscaled projections, however, show that $\sigma$ is not very sensitive to large-scale changes and less important than the changes in the mean $\overline{T}$. The discussion paper recommends two tests for checking whether the downscaling of any of the aggregated parameters are skillfully reproduced by the OLMs: by comparing their historical trends with the trends simulated by each ensemble member (a Chi-2 test) and whether their variance are realistic in terms of the simulated 90-percentile confidence interval. More details are provided in Benestad et al. (2016).

Another example demonstrating the point made by Reviewer #1 is that the number of e.g. heatwaves or storms in a region is expected to be Poisson-distributed, whereas the duration of heatwaves or dry spells may follow the geometric distribution as explained in Benestad et al. (2018). In other words, it is true there are some parameters used in the downscaling strategy that are not normally distributed, but Benestad (2021) already explains this quite extensively.

In this case, the point made by Reviewer #1 can be solved by inserting 'these' in the revised paper: "Since we downscale the parameters of the pdfs and the probability expressions, i.e. $[\mu_t, \sigma_T, f_w, \mu]$, we tend to use multiple regression because these parameters aggregated over seasonal scales tend to approximately follow the normal distribution according to the central limit theorem."

**The median**

The parameter of the geometric distribution is not an integer, but a rational number because it's the success probability $p$ and the mean spell length is $\overline{L} = 1/p$. With the mean duration, you can estimate the probabilities

for the different spell lengths, but it's impossible with the median which is not a rational number (it.s a mix between integers and sometimes a rational number). It's not common to use the geometric distribution to analyse duration of events in downscaling and regional climate modelling, and this is an original part of the downscaling approach described in Benestad (2021). Se e.g. https://en.wikipedia.org/wiki/Geometric_distribution for more information about the geometric distribution.

**The statistics of maxima**

The discussion about statistics of maxima is in the context of the IPCC SREX and presenting the change in annual maximum number of consecutive dry days. I have revised this part of the text somewhat to avoid confusion.

**Strategy for storing large volumes of multi-model ensemble ESD output**

The way huge volumes data is organised has an effect on their availability when it comes to climate services and use for climate change adaptation. There is to my knowledge no discussion about how to represent the information of large multi-model ensembles in ways so that relevant information can be quickly and efficiently distilled. This has become part of our downscaling strategy which is geared to providing regional climate change information for society, as explained in Benestad (2021). The details are provided in the reference cited: Benestad et al. (2017) and a demonstration is available from the cited URL. Reviewer #1 fails to see the direct link of this section to the other sections, which shows that (s)he has a different experience regarding the provision of regional climate information to society. One question is how (s)he would suggest to organise vast volumes of data from large multi-model ensembles represented on 8x8km maps for a given region (many gigabytes).

**RCMs and GCMs**

Yes, GCMs do strictly not represent the same aspects as those observed. But they nevertheless are able to provide useful inforation about large-scale phenomena. The downscaling strategy explained in Benestad (2021) makes use of the large-scale aspects that the GCMs are able to reproduce in order to infer local consequences.

**Using the best information the right way**

Comment: "Does this mean that the CMIP5 and CMIP6 ensemble is a 'case of not using the right information in a correct way'?" - it's better to say 'best information' and the revised paper will do that. The sentence is followed by a discussion in Benestad (2021) for why we should not use one single model, which is both obvious and a clear way of demonstrating the point. Both CMIP and CORDEX ensembles are valuable but also have limitations. We would not use any CMIP ensemble to make judgement about the fate of the Alpine snowpack, for instance, but they do a good job in terms of indicating the evolution of the global mean temperature.

**Biases in GCMs**

There are of course biases in both GCMs and RCMs, but the application of ESD bypasses them. The point is that both RCMs and ESD are needed to get the best information about the regional climate. The paper has been revised to avoid this misunderstanding.

**Wrap-up**

Benestad (2021) approaches the downscaling approach with questions why, what and how, "a golden circle" inspired by the TED-talk of Simon Sinek (https://www.ted.com/talks/simon_sinek_how_great_leaders_i nspire_action?language=en). This is not common in academic papers but it can make the points clearer for the reader. According to the paper's metric (2021-10-21), it has been accessed 780 times since July. This stands in contrast to the number of citations of the work connected to this strategy in Doblas-Reyes et al. (2021) and the attention received through CORDEX. Benestad (2021) can also be interpreted as a criticism against the protocol proposed by CORDEX and VALUE which possibly can have attracted interest.

Benestad (2021) provides a geoscientific model strategy description for using statistical models. It discusses new methods for assessment of models, including work on developing new metrics for assessing model performance and novel ways of comparing model results with observational data, in addition to describing new standard experiments for assessing model performance. Thus, the claim by Reviewer #1 that it is a review with no original model or methodological descriptions is unconvincing. Benestad (2021) also exposes controversial sides in the downscaling community, 'silo thinking,' and an intellectual gap between the different research groups (Reviewer #1's comments, DOI:10.5194/gmd-2021-176-RC1, also seems to support this interpretations). It begs the uncomfortable question whether some efforts are being suppressed by strong characters in important positions.

**R-markdown**

This is an R Markdown document. Markdown is a simple formatting syntax for authoring HTML, PDF, and MS Word documents. For more details on using R Markdown see http://rmarkdown.rstudio.com. The following chunk is provided to show details about the platform used in the demonstration above (e.g. versions of R and esd)

```
print(sessionInfo())
```

```
**R version 4.1.1 (2021-08-10)**
**Platform: x86_64-pc-linux-gnu (64-bit)**
**Running under: Ubuntu 18.04.6 LTS**
##
**Matrix products: default**
**BLAS:   /usr/lib/x86_64-linux-gnu/blas/libblas.so.3.7.1**
**LAPACK: /usr/lib/x86_64-linux-gnu/lapack/liblapack.so.3.7.1**
##
**locale:**
**[1] LC_CTYPE=en_US.UTF-8       LC_NUMERIC=C**
**[3] LC_TIME=en_US.UTF-8        LC_COLLATE=en_US.UTF-8**
**[5] LC_MONETARY=en_US.UTF-8    LC_MESSAGES=en_US.UTF-8**
**[7] LC_PAPER=en_US.UTF-8       LC_NAME=C**
**[9] LC_ADDRESS=C               LC_TELEPHONE=C**
**[11] LC_MEASUREMENT=en_US.UTF-8 LC_IDENTIFICATION=C**
##
**attached base packages:**
**[1] stats     graphics  grDevices utils     datasets  methods   base**
##
**other attached packages:**
**[1] esd_1.9.88 zoo_1.8-9  ncdf4_1.17**
##
**loaded via a namespace (and not attached):**
**[1] lattice_0.20-45  digest_0.6.27    grid_4.1.1       magrittr_2.0.1**
**[5] evaluate_0.14    highr_0.9        rlang_0.4.11     stringi_1.6.2**
```

```
**[9] rmarkdown_2.8    tools_4.1.1      stringr_1.4.0    xfun_0.23**
**[13] yaml_2.2.1       compiler_4.1.1   htmltools_0.5.1.1 knitr_1.33**
```

Benestad, R.E. 2011. *Journal of Climate* 6 (NA): 2080–98. https://doi.org/10.1175/2010JCLI3687.1.

———. 2021. "A Norwegian Approach to Downscaling." *GMD*. Copernicus. https://doi.org/10.5194/gmd-2021-176.

Benestad, R.E., B. van Oort, F. Justino, F. Stordal, K.M. Parding, A. Mezghani, H.B. Erlandsen, J. Sillmann, and M.E. Pereira-Flores. 2018. "Downscaling Probability of Long Heatwaves Based on Seasonal Mean Daily Maximum Temperatures." *Adv. Stat. Clim. Meteorol. Oceanogr.* 4 (4): 37–52. https://doi.org/10.5194/ascmo-4-37-2018.

Benestad, R.E., K. Parding, K. Isaksen, and A. Mezghani. 2016. "Climate Change and Projections for the Barents Region - What Is Expected to Change and What Will Stay the Same?" *ERL* NA (NA): NA. https://doi.org/10.1088/1748-9326/11/5/054017.

Benestad, R.E., K.M. Parding, A. Dobler, and A. Mezghani. 2017. "A Strategy to Effectively Make Use of Large Volumes of Climate Data for Climate Change Adaptation." *Climate Services* 6 (NA): 48–54. https://doi.org/10.1016/j.cliser.2017.06.013.

Benestad, R.E., K.M. Parding, H.B. Erlandsen, and A. Mezghani. 2019. "A Simple Equation to Study Changes in Rainfall Statistics." *Environ. Res. Lett.* 14 (8): NA. https://doi.org/10.1088/1748-9326/ab2bb2.

Doblas-Reyes et al., et al. 2021. "Linking Global to Regional Climate Change." Cambridge University Press. https://doi.org/NA.

---

## Author Comment (AC2)

**Response to anonymous Reviewer #2.**

I want to thank anonymous Reviewer #2 (DOI: 10.5194/gmd-2021-176-RC2) for the frank and critical comment on my manuscript (Rasmus E. Benestad 2021). Reviewer #2 thinks it "*is filled with rhetoric and subjective statements, and, surprisingly, does not contain any new scientific advancement*", which is quite different to the opinion of Reviewer #1: reading "*like a project report or a report of a laboratory for an external evaluation*". These are of course very subjective opinions and none of these descriptions are of course true. And exactly what is meant by rhetoric and what is wrong with it? (rhetoric is always present, even in the reviewer's comments)

Benestad (2021) presents a downscaling strategy on an overarching level and cites past papers and work that support the choices made for this strategy. Each cited paper can be thought of as small Lego blocks, and the strategy is the final structure containing all these little building blocks. Such an overarching strategy paper, as far as I know, is not so common in academia. It is perhaps a novel way of presenting a comprehensive approach and may be a reason for prompting two such wildly different impressions.

As is explained to Reviewer #1, very little of this work has made any impression in the international downscaling circles, which is a reason for why it's important to write a paper like this. The described strategy for downscaling has evolved over time, and the reason why we use it is exactly because we think it is superior - if it weren't so, we would of course have chosen a different one. The fact that it differs from approaches adopted elsewhere[1] also implies some criticism of those when we discuss differences. This is an expected part of the scientific discourse and part of the scientific debate. We welcome any critical view and arguments on our strategy.

The paper does not try to pretend that the whole scientific community in Norway is tied to the work described herein, and hence the title 'A Norwegian Strategy to Downscaling'. There is no reason to advance such a misconception. Nevertheless, the title is appropriate since it also describes the strategy adopted by the Norwegian Climate Services and presented in the recent Norwegian national climate reports. One of the aspects that distinguishes Norway from many other countries in terms of downscaling is that we combine both dynamical and empirical-statistical downscaling. We also try to foster a common platform for networking and collaborations: https://sites.google.com/met.no/downscaling. But is this an important issue with this paper? I urge Reviewer #2 to please explain exactly what is rather contradictory and worrisome with it and its title, because this is an unexpected comment which is hard to interpret. For me, this concern seems really far-fetched.

**Why this paper?**

The world is experiencing rapid climate change and urgently needs to start climate change adaptation, which needs to be based on the best information that we can provide on future risks and opportunities. There is already a motion prompted by the Paris 2015 accord, the Climate Adaptation summit, Copernicus C3S, and the upcoming COP26 (2021). My group has
* * *
[1] I think, but I haven't seen any similar paper outlining the general thinking on downscaling and the provision of regional climate information for society.

long experience with climate analysis and downscaling, geared towards climate change adaptation, but my impression is that our progress has until now been ignored outside Norway, as explained to Reviewer #1. Hence, it's important to share our experience, and rather than getting trapped in a forest of details, it's better to take a bird's eye overview to convey the rationale behind the downscaling strategy. All the details are of course available from the cited papers (if some of them are not open-access, please let me know).

**What is Mainstream?**

Good question, and perhaps there could be a better term for it. In this context, it is the norm within the downscaling community, both for ESD and for RCMs - in addition to the attitudes and beliefs shared within e.g. WCRP, CORDEX and COST-Value. The chapter on regional climate modelling in the recent IPCC AR6 WG1 (2021) reflects some of them. It is true that many countries have their own regional climate simulations with spatial and temporal resolutions that far exceed those of CORDEX. Nevertheless, CORDEX also involves more than spatial and temporal resolutions. For instance, Benestad (2021) refers to protocols for evaluation of the methods. In the revised paper, a definition will be given for the term mainstream: "*The term 'mainstream' is used here when referring to the most common norms within the international circles of downscaling, such as the protocols adopted by CORDEX and COST-Value, however, this notion may not necessarily be acknowledged by everyone since there are many differences between individual research groups*".

**Common EOFs**

Reviewer #2 makes a good point that others may use a different term than 'common EOFs', but doesn't suggest what these terms might be. Nevertheless, a read through the IPCC AR6 WG1 on regional climate modelling reveals that common EOFs, or the same concept under different names, are not present in the assessment of the science on regional climate. My own experience is that people often don't understand the concept, and I therefore challenge Reviewer #2 to find more than 63 papers on downscaling that are based on common EOFs - otherwise the claim that "the reasoning behind l. 80 is flawed" is misleading. This particular remark is also a bit weird (and rhetorical), since sentence in question merely observes a fact: "*In spite of the success with utilising common EOFs, a Google scholar search on '"common EOFs" downscaling' only had 63 hits (of which about 40 referred to our own work), despite more than 20 years since they first were introduced in ESD and the widespread need for climate change adaptation and downscaled results.*" Reviewer #1 offered a different opinion on this finding: that our colleagues don't believe the common EOFs have any merit. Nevertheless, this doesn't disqualify a discussion about their merit and appearance. The point with this remark was that the common EOFs appear not to be widely used and we think they are highly underrated. This point is worth sharing within the science community.

**Equations**

The paper really is about the downscaling strategy, which is a level above the equations. Hence the comment about equations is a bit irrelevant here. The equations themselves are provided in the cited literature with mostly open-access papers (please let me know if they aren't). The headings of sections 2.1 – 2.6 are '*Main differences to the mainstream*', '*Reasons for why a different approaches*', '*Different choices*', '*"Downscaling climate" approach*', '*Common protocols concerning downscaling*', and '*ESD is suitable for downscaling extremes*' - it's not unnatural that they are mostly qualitative. They do, however, cite works which demonstrate the merit of the different choices behind the chosen strategy. There will also be some degree of subjective choices, as there always is in science. When it comes to the "best way", the paper presents nine criteria for evaluating the results, one of them being the use of common EOFs that also provide a quality control on the large-scale conditions simulated by GCMs. The paper argues that this is an improvement over not including such a quality control. This whole comment is a bit surprising, as it seems to disqualify both machine learning and artificial intelligence.

**Details of approach**

Moreover, the paper really focuses on the overarching strategy on a higher level, rather than lower-level technicalities. The comment "*precise details of his approach in the paper*" is therefore a bit out of place, and suggests that the novel approach of this paper may be confusing for readers with a conserative set of expectations. The paper should be fairly clear about its objective, and the first sentence of the abstract is: "*A description of a comprehensive geoscientific downscaling model strategy is presented outlining an approach that has evolved over the last 20 years*". I struggle to find a clearer way to say it, sorry! This comment, combined with the observation that there have been 740 views of (Benestad 2021) since July, gives a clear indication for the need of such a paper - because there seems to be intellectual barriers between colleagues in the downscaling community.

I must admit that I don't understand the following sentences: '*It is even unclear whether the reported results appear elsewhere. So, what is the purpose of peer review?*' - how do they fit into this context? When the complaints above are about citing previous papers, then the objections seem to be very thin. Critical comments should be based on real substance, and the reviewer has not even discussed the actual strategy presented in (Benestad 2021) - just weird aspects.

**Probability distribution functions**

The point about describing and explaining the time dependence of pdf parameters is useful and such an explanation will be provided in the revised paper: "*This approach seeks the dependency of the pdf parameters representing local climate statistics on the large scales, and in practise it involves aggregating the parameters on a seasonal or annual basis. Thus, we end up with a shorter time series of such parameters, which are then used as predictand variables in the downscaling methods against large-scale predictors aggregated over the same timescales.*"

The original paper does offer a discussion on spatial dependence which is ensured by PCA - it already states this: "*We use PCA to represent the predictands because they inherently ensure the same spatial covariance as seen in the observations, in addition to reorganising the data to emphasise the large-scale variability* (Benestad et al. 2015)". We have not yet included more than one variable statistics, but it is possible to let the PCA represent both temperature and rainfall statistics - the maths is the same.

**Comparison with RCMs.**

For comparison with RCMs, the paper cites (Mezghani et al. 2019) and (Hanssen-Bauer et al. 2009), but there is also some grey literature (Met Norway reports) and some presentations in the Norwegian Climate Services. This covers Norway as well as Poland (Mezghani et al. 2019), but has been limited to specific projects and available funding. It would be great to extend this work to other places too, but we have limited resources and time. It is our hope that by sharing our downscaling strategy, with the help of the open-source `esd` (available from https://github.com/metno/esd), our colleagues will become interested and try to reproduce this type of analysis elsewhere independent of us. This hopefully can happen more easily with this paper.

**Extremes**

It is true that when qq-plots deviate from the 1-1 line it shows that the Gaussian distribution is unsuitable for extremes. A careful read of section 2.6 '*ESD is suitable for downscaling extremes*' conveys that the Gaussian distribution **is not used** for extremes, but ESD can deal with them in a different fashion that nevertheless is well-suited for extremes: in terms of how often e.g. heatwaves occur, assuming Poisson process with low probability, or how long they last, assuming that the duration follow a geometric distribution with a particular success probability p. The original paper explains it this way: "*To estimate the probability of long-lasting events, the geometric distribution needs to be combined with the probability of the events occurring (e.g. Bayesian approach where the probability of the event can be modelled as a Poisson process)*". More details are provided in Benestad et al. (2018), but there are only a few scientific papers on use of statistics in the field of climate research (cited in the cited papers).

**Specifics**

- I don't understand what is meant by a biased *statement of "state-of-the-art", sorry*!
- *'The goal of the paper is not stated'* - the abstract says it is to present a description of a comprehensive geoscientific downscaling model strategy.
- 'New weather-related hazards' are those that come in the future. Some may even be a new type for a given region. This comment is rhetorical.
- The paper does not aim to present an exhaustive list of types of downscaling approaches. Hopefully, other research groups will present their strategy and compare it with others/the norm/what-ever.

- "Mainstream" will be defined in the revised version.
- In this context, linear algebra involves EOFs and PCA in addition to vectors holding the data and ways of dealing regression on computers. There is a book with the title 'Linear algebra' (Strang 1988) that provides a good background.
- No problem that Reviewer #2 doubts that CORDEX can be considered as "mainstream" for downscaling. This is a minor issue.
- The passage "*and some will say it gives the right answers for the wrong reasons*" will be dropped in the revised version.
- Modelling temperature by a Gaussian distribution is not advocate for the extremes. See the point on extremes above.
- The question whether the variables are iid - it depends. There is little year-to-year dependence between the parameter estimates, as explained in the original paper. Within a season, there is autocorrelation for temperature, which means that the real degrees of freedom for a season are less than 90. Still, testing them for normality (qq-plot) tends to give reasonable results, and the mean value is also mostly non-problematic. For all intents and purposes, the results are useful for society.
- Modelling the precipitation by an exponential distribution will be more carefully explained in the revised paper: "*It is well-known that 24-hr precipitation doesn't follow the exponential distribution, and especially for the more extreme rainfall amounts. It can nevertheless provide a useful framework for the analysis of 24-hr rainfall (Benestad et al. 2012a; 2012b) and by introducing a 'scaling factor' into this framework, it is possible to to get a more accurate representation (Benestad et al. 2019).*"
- The details about the way the parameters of the statistical laws are "downscaled" are provided in the cited papers, but also explained in this paper during the discussion of the use of regression on aggregated parameters (this prompted som comments from Reviewer #1). Also, more details will be provided in the revised paper.
- Extremes of temperature and precipitation ***are not*** modelled with any assumption of the Gaussian distribution, and for 24-hr rainfall, the exponential distribution is merely used as a starting point (a kind of framework) with a modification to capture the deviations from it. These details are discussed in the cited papers. Benestad (2021) should be fairly clear on these points if the manuscript is read carefully.
- The point regarding '*The strategy of "keeping things simple and elegant (mathematical)"*' referred to the merits of using common EOFs and downscaling the parameters of the pdfs directly, but there is still a need for thorough evaluation (nine levels). The remark of *"spherical cows in vacuum"* is, using Reviewer 2's own words, merely *rhetorical.*
- GCM grids are the grid meshes on which the GCMs represent variables such as temperature and rainfall. Isn't that obvious?
- Common EOFs deal with differences in mean values by combining respective anomalies from reanalyses and GCM results, and the mean climatology from the station observations is subsequently added to the output of the ESD results to provide a downscaled record. If PCA are used as predictand, then the PCA is estimated on the anomalies of the station data, the PCA is projected through ESD, and results corresponding to the original data are derived after the PCA have been 'reversed' to recover the original data structure with the mean values subsequently added. Keeping

all these details in this overview paper would get any reader lost in the forest of details, but they are available from the cited literature. And again, the fact that I need to explain how common EOFs are used may suggest that the choices in our strategy, for instance using common EOFs and PCA, are unfamiliar for most of our colleagues. This is a bit ironic, since this question suggests that the use of common EOFs are not so common and that the results from the Scholar Google search actually do reflect the situation.

- The strategy for storing large multivariate data also involves common EOFs, and it may be a bit tricky to explain to people who are not familiar with them. It's explained in the cited paper (Benestad et al. 2017). The point with this section is also explained to Reviewer #1: It deals with concerns of making regional climate information accessible for the society.
- It is true that Figure 6 does not demonstrate anything on the storage method - but it only took seconds to produce it from a large multi-model ensemble. Try the cited URL instead.
- *What makes the author believe of "silo thinking"?* Separate CORDEX white papers for RCMs and ESD, for instance.

**References**

Benestad, Rasmus E. 2021. "A Norwegian Approach to Downscaling." Preprint. Climate and Earth system modeling. https://doi.org/10.5194/gmd-2021-176.

Benestad, Rasmus E., Deliang Chen, Abdelkader Mezghani, Lijun Fan, and Kajsa Parding. 2015. "On Using Principal Components to Represent Stations in Empirical-Statistical Downscaling." *Tellus A* 67 (0). https://doi.org/10.3402/tellusa.v67.28326.

Benestad, Rasmus E., Bob van Oort, Flavio Justino, Frode Stordal, Kajsa M. Parding, Abdelkader Mezghani, Helene B. Erlandsen, Jana Sillmann, and Milton E. Pereira-Flores. 2018. "Downscaling Probability of Long Heatwaves Based on Seasonal Mean Daily Maximum Temperatures." *Advances in Statistical Climatology, Meteorology and Oceanography* 4 (1/2): 37–52. https://doi.org/10.5194/ascmo-4-37-2018.

Benestad, Rasmus E, Kajsa M Parding, Helene B Erlandsen, and Abdelkader Mezghani. 2019. "A Simple Equation to Study Changes in Rainfall Statistics." *Environmental Research Letters* 14 (8): 084017. https://doi.org/10.1088/1748-9326/ab2bb2.

Benestad, Rasmus, Kajsa Parding, Andreas Dobler, and Abdelkader Mezghani. 2017. "A Strategy to Effectively Make Use of Large Volumes of Climate Data for Climate Change Adaptation." *Climate Services*, June. https://doi.org/10.1016/j.cliser.2017.06.013.

Benestad, R.E., D. Nychka, and L. O. Mearns. 2012a. "Spatially and Temporally Consistent Prediction of Heavy Precipitation from Mean Values." *Nature Climate Change* 2 (doi: 10.1038/NCLIMATE1497): 544–47.

———. 2012b. "Specification of Wet-Day Daily Rainfall Quantiles from the Mean Value." *Tellus A* 64 (14981). https://doi.org/10.3402/tellusa.v64i0.14981.

Hanssen-Bauer, I., H. Drange, E. J. Førland, L. A. Roald, K. Y. Børsheim, H. Hisdal, D. Lawrence, et al. 2009. "Klima i Norge 2100." september 2009. Oslo: Norsk klimasenter.

Mezghani, Abdelkader, Andreas Dobler, Rasmus Benestad, Jan Erik Haugen, Kajsa M. Parding, Mikołaj Piniewski, and Zbigniew W. Kundzewicz. 2019. "Sub-Sampling Impact on the Climate Change Signal over Poland Based on Simulations from Statistical and Dynamical Downscaling." *Journal of Applied Meteorology and Climatology*, March. https://doi.org/10.1175/JAMC-D-18-0179.1.

Strang, G. 1988. *Linear Algebra and Its Application*. San Diego, California, USA: Harcourt Brace

& Company.